# Hydrophilic Interaction Liquid Chromatography Coupled to Mass Spectrometry and Multivariate Analysis of the *De Novo* Pyrimidine Pathway Metabolites

**DOI:** 10.3390/biom9080328

**Published:** 2019-07-31

**Authors:** Paula Galeano Garcia, Barbara H. Zimmermann, Chiara Carazzone

**Affiliations:** 1Laboratory of Advanced Analytical Techniques in Natural Products, Universidad de los Andes, Bogotá 111711, Colombia; 2“Grupo de Investigación en Productos Naturales Amazónicos”, Facultad de Ciencias Básicas, Universidad de la Amazonia, Florencia 180002, Colombia; 3Departamento de Ciencias Biológicas, Universidad de los Andes, Bogotá 111711, Colombia

**Keywords:** hydrophilic interaction liquid chromatography, zwitterionic column, experimental design, plant-pathogen interaction, Box–Behnken design, central composite design, tomato plants, *Phytophthora infestans*

## Abstract

In this study, we describe the optimization of a Hydrophilic Interaction Liquid Chromatography coupled to mass spectrometry (HILIC-MS) method for the evaluation of 14 metabolites related to the *de novo* synthesis of pyrimidines (*dn*SP) while using multivariate analysis, which is the metabolic pathway for pyrimidine nucleotide production. A multivariate design was used to set the conditions of the column temperature, flow of the mobile phase, additive concentration, gradient rate, and pH of the mobile phase in order to attain higher peak resolution and ionization efficiency in shorter analysis times. The optimization process was carried out while using factorial fractional designs, Box–Behnken design and central composite design while using two zwitterionic columns, ZIC-p-HILIC and ZIC-HILIC, polymeric, and silica-based columns, respectively. The factors were evaluated while using resolution (R), retention factor (*k*), efficiency of the column (N), and peak height (h) as the response variables. The best optimized conditions were found with the ZIC-p-HILIC column: elution gradient rate 2 min, pH 7.0, temperature 45 °C, mobile phase flow of 0.35 mL min^−1^, and additive (ammonium acetate) concentration of 6 mM. The total analysis time was 28 min. The ZIC-p-HILIC LC-MS method yielded satisfactory results for linearity of calibration curves, limit of detection (LOD), and limit of quantification (LOQ). The method has been shown to be appropriate for the analysis of *dn*SP on samples of tomato plants that were infected with *Phytophthora infestans*.

## 1. Introduction

Purine and pyrimidine nucleotides are important metabolites that are involved in many processes of living organisms. Nucleotides are the building blocks in the synthesis of nucleic acids, and they are the precursors of primary and secondary metabolites [1]. The alteration in levels of the metabolites that are related to or involved in the *de novo* synthesis of pyrimidines (*dn*SP) offers information regarding the response to different stimuli that might occur in the organism [2]. Particularly, in plant-pathogen interactions, phytopathogens, as parasites, are obliged to obtain nutrients from existing sources, invading and adapting to the host tissue to eventually reproduce [3]. In many pathogens, the partial or complete gain or loss of genes that are related to nucleotide metabolism are closely correlated with their life cycles and metabolic interactions with the hosts [2], so studying the metabolism of *dn*SP compounds can help to elucidate the plant-pathogen interaction.

The *dn*SP metabolites belong to different chemical families that include amino acids, nitrogen-containing bases, nucleosides, mono-, di-, and triphosphate nucleotides, nucleotide precursors and derivatives. They contain several hydroxyl, carbonyl, and amino groups, and they thus have highly different degrees of polarity (Appendix A) [4,5,6], causing low retention on reverse phase columns, mainly with the use of polar aqueous-organic mobile phases. Therefore, chromatographic resolution is low in these methodologies and it presents many co-elution phenomena [7]. In this context, Hydrophilic Interaction Liquid Chromatography (HILIC) has established itself as the most appropriate chromatographic mode for the highest retention and the best separation of highly polar and hydrophilic compounds [8,9].

HILIC is characterized by a polar stationary phase and a polar aqueous-organic mobile phase that contains a high proportion of organic modifier, which is usually acetonitrile. This highly volatile organic mobile phase provides low back pressure due to its low viscosity. In addition, it promotes the formation of smaller droplets due to a decrease in the surface tension of the water and, consequently, greater desolvation efficiency, giving rise to a significant increase in the number of ions that formed in the electrospray source (ESI) [8,9,10,11,12,13]. Alpert suggests that the partition between the aqueous phase that is associated with the stationary phase and the organic component of the mobile phase is the main retention mechanism in HILIC, whereas the retention in normal phase liquid chromatography (NPLC) is dominated by surface adsorption phenomena [14,15,16]. However, the participation of secondary interactions (dipole-dipole, hydrogen bonding, and ion exchange) may play an important role in the separation, leading to changes in selectivity [17,18,19,20]. These "secondary" interactions significantly contribute to the separation on HILIC depending on the nature of the individual components of the experiment, such as stationary phase, mobile phase, analyte, and the interactions between them [20,21]. The design of experiment (DoE) was used to study the effects of the variation of several chromatographic conditions on the retention and ionization of the compounds on HILIC while using liquid chromatography coupled to mass spectrometry with electrospray ionization (LC-ESI-MS) in order to develop an efficient method for the separation and detection of *dn*SP metabolites. Two zwitterionic columns were used to test a broad pH range—a polymeric ZIC-p-HILIC and silica-based ZIC-HILIC columns. The use of the DoE principles allowed for the simultaneous study of multiple factors, such as column temperature, mobile phase flow, additive concentration, gradient rate, and mobile phase pH to achieve the higher responses on resolution, column efficiency, peak height, and retention factor, with a significant decrease in the number of experiments. This approach not only helped to shorten the time needed to complete the study, but it also provided a better understanding of the relative importance and interrelation of the experimental results [21,22,23]. The developed methodology was applied to the *dn*SP metabolites analysis in tomato plants infected with *Phytophthora infestans,* which is one of the most devastating tomato diseases, demanding high chemical input for worldwide disease control [24,25,26,27].

## 2. Materials and Methods

### 2.1. Chemicals and Reagents

The standard orotic acid, dihydroorotic acid (DHO), cytidine, uridine, uracil, aspartic acid, glutamine, carbamoyl aspartic acid, uridine diphosphate glucose (UDP-Glu), uridine monophosphate (UMP), cytidine triphosphate (CTP), uridine triphosphate (UTP), adenosine triphosphate (ATP), and guanidine triphosphate (GTP) were purchased from Sigma-Aldrich (St. Louis, MO, USA). Appendix A summarizes the characteristics of each compound. The HPLC grade reagents ammonium acetate, ammonium formate, acetic acid, formic acid, and ammonium hydroxide were purchased from Sigma-Aldrich. The HPLC grade solvents, acetonitrile, and methanol, were purchased from Honeywell International, Inc. (Morris Plains, NJ, USA).

### 2.2. Apparatus

All of the experiments were carried out on a 1260 Infinity UHPLC coupled to an Agilent 6250 iFunnel quadrupole Time of Flight Q-TOF LC-MS system with Agilent Dual Jet Stream technology of the electrospray ionization (ESI, Agilent Technologies, Santa Clara, CA, USA). 2.1 × 150 mm (Merck SeQuant, Umeå, Sweden) columns were used for the separation the ZIC-HILIC 3.5 µm, 2.1 × 150 mm (Merck SeQuant, Umeå, Sweden) and ZIC-p-HILIC 5 µm. The temperature was varied according to the experimental design. The mobile phases were A (aqueous ammonium acetate or ammonium formate) and B (acetonitrile). Phase B switched from 95% to 40% with a gradient rate that is denominated 2, 3, and 4 (indicating to switch B from 95% to 40% in 28, 18, and 14 min, respectively) and was maintained for 5 min at 40% with a flow according to the experimental design. Subsequently, the initial conditions were reached in 2 min and the column was equilibrated for “**X**” min, depending on the flow used (0.25 mL min^−1^, X: 15 min; 0.20 mL min^−1^, X: 18 min; and, 0.15 mL min^−1^, X: 24 min) (Appendix A). The injection volume was 2 µL.

### 2.3. Stock Solution Preparation

GTP, ATP, aspartic acid, uracil, orotic acid, and glutamine stock solutions were prepared in water and UMP, UDP-Glu, UTP, CTP, DHO, cytidine, uridine, and carbamoyl aspartic acid in methanol: water (1:1). All of the solutions were prepared at 1 mg mL^−1^ and stored at −20 °C. The working solutions were prepared from stock solutions. All of the solutions were stored at −20 °C for one month.

### 2.4. Sample Preparation

#### 2.4.1. Tomato Plants

Tomato seeds were obtained from the local market and then germinated in germination chamber (120 mm Petri dishes containing wet germination paper) that were maintained at 18 °C for 16 h of photoperiod for three days. Subsequently, the germinated seeds were planted in 16 cm diameter pots containing 1:1 mixtures of soil and vermiculite and subsequently sub-irrigated once a day. The plants were maintained at 18 °C, 16 h photoperiod, and around 60–70 % relative humidity (RH) in a greenhouse.

#### 2.4.2. Phytophthora infestans Strain

A pure culture of *P. infestans* was sub-cultured on V8 media [28,29] in 90 mm Petri dishes at 18 °C. After 2–3 weeks, a sporangial suspension was prepared by scraping the surface of the colonies with a sterile scalpel and the mycelia were suspended in sterilized water to produce the infecting solution. The concentration of sporangia in the suspension was adjusted to 1.0 × 10^5^ sporangia mL^−1^ while using a Neubauer chamber.

#### 2.4.3. Plant Inoculation

After 5–6 weeks of growth, the tomato plants were inoculated with 10 μL of the sporangial suspension of *P. infestans* at four different sites, two on each side of the midrib of the leaf. 96 h post inoculation, infected tomato leaves were excised and then immediately macerated in liquid nitrogen. 100 mg of the crushed powder were vortexed (Multi Reax, Heidolph, Germany), mixed with 1 mL of methanol for 10 min at room temperature, and then centrifuged for 5 min at 12,000× *g* at 20 °C (Centrifuge 5418, Eppendorf, Hamburg, Germany). The supernatants were stored at −20 °C until analysis [29].

### 2.5. Multivariate Design

Multivariate design was employed as a strategy to attain higher peak resolution and ionization efficiency in a shorter analysis time. The optimization of the chromatographic conditions for each HILIC column was firstly carried out by fractional factorial design (2^5−2^), for the initial selection of the significant factors before an optimization of the critical conditions. Table 1 details the minimum, average, and maximum levels of each variable. The factors, temperature of the column, flow of the mobile phase, additive concentration, gradient rate, and pH of the mobile phase were evaluated using resolution (R), retention factor (*k*), efficiency of the column (N), and peak height (h) as the response variables.

The statistical calculations of the p value for the significance of the effect values of the variables were made using Statistica v6.0 package (Statsoft, Tulsa, OK, USA) [22,23].

### 2.6. Analytical Parameters

For linearity, the calibration curves of each standard were performed while using different concentration ranges (three replicates of each point at eight concentrations). The sensitivity of the method was evaluated by determining the limit of detection (LOD) and the limit of quantification (LOQ), being expressed as the concentration calculated at a signal-to-noise ratio (S/N) of 3 and 10, respectively. The matrix effect was evaluated to ensure that the extract did not influence the quantification of the compounds. The extracts of *P. infestans* infected tomato plants were used at a concentration of 50 mg L^−1^ and three different concentrations of standards mix were added (low, medium, and high concentrations).

The infected tomato plant extracts were also analyzed at 50 mM to establish the concentration of *dn*SP metabolites.

## 3. Results and Discussion

### 3.1. Factorial Fractional Designs

The chromatographic optimization was carried out while using multivariate strategies with factorial design (2^5−2^), using the efficiency (N), the retention factor (*k*), the resolution (R), and the peak height (h) as response in order to evaluate the factors listed in Table 1.

The first factorial fractional design using a ZIC-p-HILIC column was carried out while using 2^5−2^ design with three replicates in the central point, which results in a total of 19 experiments. For the efficiency (N), the results showed that T, Conc, Flow, and Grad were significant with a 95% confidence level (Pareto’s chart, Figure 1a). Positive values for T (12.71), Conc (9.25), and Grad (6.92) indicated that increasing these parameters would result in response increment, and hence in better results. A negative value for Flow (−7.82) meant that decreasing the flow rate would also contribute to an improved response. pH was not statistically significant at the 95% confidence level, so that any value in the studied range could be used. For the retention factor (*k*), the results showed that T, Conc, Flow, Grad, and pH were significant at the 95% confidence level (Pareto’s chart, Figure 1b). Positive values for Flow (107.44), Conc (36.25), T (32.70), and pH (30.55) indicated that their increase could result in enhanced responses. A negative value for Grad (−110.14) meant that decreasing the gradient rate would also contribute to an improved response. For the resolution (R), the results showed that T, Conc, Flow, Grad, and pH were significant at the 95% confidence level (Pareto’s chart, Figure 1c). Positive values for Flow (16.59), Conc (32.74), T (48.52), and pH (7.30) indicated that their increment could result in increased responses. A negative value for Grad (−47.89) meant that a decreasing gradient rate would also contribute to an improved response. Finally, concerning the peak height (h), the results showed that T, Conc, Flow, and pH were significant with a 95% confidence level (Pareto’s chart, Figure 1d). Positive values for T (7.59), pH (12.35), and Flow (4.42) indicated that increasing these parameters would result in a response increment, and hence in better results. A negative value for Conc (−9.45) meant that decreasing the additive concentration would also contribute to an improved response. Grad was not statistically significant with a 95% confidence level, so we could use any value in the studied range.

Similarly, the second factorial fractional design for the ZIC-HILIC column was carried out while using 2^5−2^ design with three replicates in the central point, resulting in 19 experiments. For the efficiency (N), the results showed that pH and T were significant with a confidence level of 95% (Pareto’s chart, Appendix A): a positive value for pH (17.26) and a negative value for T (−7.80). Flow, Conc, and Grad were not statistically significant at 95% confidence level. For the retention factor (*k*), the results showed that T, Flow, Grad, and pH were significant with a confidence level of 95% (Pareto’s chart, Appendix A). Positive values were obtained for Flow (13.84) and pH (56.69) and negative values for Grad (−41.24) and T (−12.64). Conc was not statistically significant at the 95% confidence level. In the case of resolution (R), the results showed that T and Grad were significant at the 95% confidence level (Pareto’s chart, Appendix A), giving a positive value for T (5.01) and a negative value for Grad (−7.05). Flow, Conc, and pH were not statistically significant with a confidence level of 95%. Finally, for the height (h), the results showed that Flow, Conc, Flow, and pH were significant at the 95% confidence level (Pareto’s chart, Appendix A). Positive values were obtained for pH (14.25) and Flow (7.80) and a negative value for Conc (−8.89). Grad and T were not statistically significant at the 95% confidence level.

Using the polymeric column (ZIC-p-HILIC), the increase in the additive concentration enhanced the retention factor (*k*). Arase and collaborators [30], who developed monophosphate nucleotide analysis in zwitterionic ZIC-HILIC columns through DoE, report similar results. The *k* increased with the increase in ammonium acetate between 0.5–15 mM, however it decreased at higher concentrations. This is attributed to a greater water adsorption on the HILIC surface. However, the large ionic strength prevents the ion exchange between the stationary phase and the analyte at concentrations higher than 15 mM. Likewise, an increase in *k* is reported by augmenting the content of ammonium acetate to approximately 200 mM, after which the retention slightly decreased for the separation of mono, di-, and triphosphate nucleotides through a FRULIC-N column [31]. The increase retention is related to the higher concentration of solvated ions in the adsorbed aqueous layer, obtaining a thicker adsorbed water layer and a higher retention [32].

The experiments were carried out while varying the type of buffer/pH from 4 to 7 and 3 to 6 for ZIC-p-HILIC and ZIC-HILIC, respectively, showing that an increase in pH caused an increase in the retention of the compounds on both of the columns. Similar results are reported for the retention of nucleotides mono-, di-, and triphosphate in a FRULIC-N column [31]. Higher *k* values for this group of compounds are related to the pK_a_s for the first and second protons of the phosphoric acid residue, that are ~1 or less, and ~6–7, respectively. Therefore, the net charge in the nucleotide is −1 at neutral or lower pH. At a low pH, the interaction of the monovalent anion with the stationary phase is relatively low when compared to a high pH. This is probably the reason for the low retention factors that were observed for the nucleoside mono-, di-, and triphosphates at low pH. Although at increasing pH, the phosphate groups acquire their second negative charge and the amino groups in the nitrogen bases are also neutralized [4,31,33]. The analysis showed that the compounds retention is favored by the decrease in the gradient rate, both on ZIC-p-HILIC and ZIC-HILIC; Arase report similar results [30].

On the other hand, the ZIC-p-HILIC column efficiency (N) increased with the additive concentration. Similar results have been reported for the separation of nucleoside triphosphates on a ZIC-p-HILIC, suggesting that high concentrations of the buffer produce greater efficiency of the column (N), although longer retention times [34]. Alpert, Johnsen, and colleagues [34,35] suggest that the increase in retention time with salt concentration is probably due to the shielding of negatively charged sites, which decrease electrostatic repulsion. In addition, the number of theoretical plates per meter of each compound was low at a lower concentration of ammonium acetate, which indicated a stronger ionic interaction between the stationary phase and the analyte under these conditions [30].

Our results showed that N increased with a temperature decrease in ZIC-HILIC, but with a temperature increase in the ZIC-p-HILIC. Johnsen and colleagues [34] also indicate higher efficiencies on the ZIC-p-HILIC column are obtained with high temperature and emphasize that the retention time also increases with the increase of temperature. Contrary to what occurs in RPLC, where the increase in temperature generally leads to a change in the equilibrium towards the mobile phase, which shortens the retention time [36,37]. It could be speculated that the temperature affects the hydration of the analytes to explain this behavior in polymeric zwitterionic columns, therefore affecting retention [38].

Regarding the resolution, our results indicated that a slower gradient rate increased the resolution of *dn*SP metabolites. However, lower gradient rate values corresponded to longer elution times (28 min). As the content of organic solvent decreases, the significance of the various interactions that are involved in HILIC may vary [34]. Additionally, the results that were obtained in the factorial designs 1 and 2 showed that the increase in temperature allows for an increase in the resolution in both ZIC-p- and ZIC-HILIC columns. Arase et al. [30] report similar results in monophosphate nucleotides, for which R increases with long gradient elution times and high temperatures.

Finally, the peak height (h) and the area of the peaks are related to the ionization efficiency of the compounds under the conditions that were tested. The pH is the most significant variable in both columns. Therefore, these results indicate that most compounds that were analyzed by HILIC will have a higher mass-to-charge signal when they are ionized (high pH for acids) [39]. Additionally, the high organic content in the mobile phase leads to a rapid evaporation of the solvent during the ionization by ESI, which increases the ionization of the compounds and the sensitivity of the analysis [9].

### 3.2. Box-Behnken Design and Central Composite Design

The factorial fractional design allows for reducing the number of significant variables from five to three for the ZIC-p-HILIC column (Appendix A) and being close to ideal, Box–Behnken design was performed to estimate the response surface to find its maxima (that correspond to critical points or optima). Similarly, the central composite design was performed as the number of significant variables for ZIC-HILIC was reduced to two (Appendix A). In Table 2, factors and levels used in the construction of the Box–Behnken and central composite designs are described for ZIC-p-HILIC and ZIC-HILIC, respectively.

The Box-Behnken experimental design was used in association with response surface methodology for the chromatographic optimization on the ZIC-p-HILIC column, which resulted in 18 experiments with six replicates at the central point [23].

The response surface of the retention factor (*k*) (Figure 2) reached optimal temperature values throughout the studied range (35–45 °C), but with an increasing tendency at temperatures below 35 °C or above 45 °C (Figure 2a,b). Similarly, *k* reached a maximum value for the concentration of approximately 10 mM (Figure 2b,c). Likewise, *k* tended to increase with the flow enhance and it was able to reach the maximum for values above 0.35 mL min^−1^ (Figure 2a,c).

The response surface of the resolution (R) (Figure 3) achieved higher values when the temperature reached the maximum at 40 °C (Figure 3a,b), with concentrations higher than 15 mM (Figure 3b,c) and flows below 0.25 mL min^−1^ or above 0.35 mL min^−1^ (Figure 3a,c). The flow had very little influence on the resolution, since the surface is mostly green or yellow (it reached low or medium values).

The response surface of the peak height (h) (Figure 4) tended to increase with the temperature values in the range from 38 to 42 °C (Figure 4a,b), with flows that were lower than 0.25 mL min^−1^ (Figure 4a,c) and at concentrations lower than 6 mM (Figure 4b,c).

Finally, the response surface of the column efficiency (N) (Figure 5) reached higher values when the temperature was 40 °C (Figure 5a,b), with additive concentrations that were higher than 15 mM (Figure 5a,c) and flows below 0.25 mL min^−1^ (Figure 5b,c).

In the ZIC-HILIC column, the response surface of the retention factor (*k*) (Figure 6a) tended to increase for temperature below of 20 °C and pH less than 5.6. Meanwhile, the response surface of the parameter (N) tended to increase with a temperature below 30 °C and pH less than 5.4 (Figure 6b). The response surface of the parameter (R) tended to increase with temperature below 10 °C and it remained constant throughout the pH range studied 5.2–6.8 (Figure 6c). Finally, the response surface of the parameter (h) had the tendency to increase with a temperature above 25 °C and increases with pH values above 6.8 and below 5.2 (Figure 6d).

Appendix A summarizes the optimized values of each parameter (temperature, additive concentration, and flow) of the ZIC-p-HILIC and ZIC-HILIC columns. According to the results, the values at which each factor reached optimal responses (R, N, h, and *k*) are clearly different, hence it is necessary to prioritize the response according to the interests of the analysis. When considering that *(i)* the metabolites that are related with the pyrimidine pathway have great structural and functional diversity and *(ii)* biological samples are complex matrices in which the ionic suppression of the metabolites could be increased, we chose to improve the peak height (h), which made it possible to increase sensitivity in the detection of metabolites in the samples. Based on the above criteria, a triplicate analysis was performed with the ZIC-p-HILIC and ZIC-HILIC columns in ESI (−)−QTOF-MS in order to establish the optimal values of the chromatographic analysis while using the mixture of metabolites under the following conditions: (a) gradient rate 2; pH 7.0; temperature 45 °C, flow 0.35 mL min^−1^; and, three concentrations 6, 10, and 13 mM of ammonium acetate for ZIC-p-HILIC and (b) gradient rate 2; pH 5.2; flow 0.25 mL min^−1^; additive concentration 5 mM and three temperature 10, 20, and 30 °C for ZIC-HILIC analysis. In ZIC-p-HILIC, the experiment at 6 mM enabled the highest peak height (17,518.5), that is, the better ionization of the compounds. In ZIC-HILIC, the experiment at 20 °C enabled the highest peak height (49,388.1). However, the organic polymeric column (ZIC-p-HILIC) provided a markedly improved chromatographic yield than ZIC-HILIC. Firstly, the column can be used at a higher pH, which resulted in a higher number of symmetric peaks, higher retention factors, greater separation, and greater sensitivity of *dn*SP metabolites. Secondly, the polymer column does not contain silanol residual groups, which can be a source of secondary interactions that cause tailing [14,33,34].

According to multivariate designs, the optimized chromatographic conditions for ZIC-p-HILIC were as follows: oven temperature: 45 °C; flow 0.35 mL min^−1^; pH 7.0; gradient rate 2.0; and, concentrations 6 mM of ammonium acetate. The analysis was not continued in the ZIC-HILIC column, because there was a high peak tailing and high noise in the extracted ion current (EIC). Appendix A shows *dn*SP metabolites related the EIC chromatogram, which were acquired under optimized conditions allowing for excellent separation of the 13 of the 14 *dn*SP compounds in a total analysis time of 28 min. The GTP was not detected under the optimized conditions, although the conditions were varied to analyze the 14 compounds. The EIC depicts the following elution order: nucleobase < nucleosides > acid precursors > amino acids > nucleotide monophosphate < nucleotide diphosphates < nucleotide triphosphates. According to the literature, the retention behavior of bases, nucleosides, and nucleotides, in HILIC with a high percentage of ACN 95% in the mobile phase, is governed by hydrophilicity, and the retention factors increase with the polarity of the solute and molecular size [4,35,40]. In addition, it is already expected that an increase in the number of phosphate groups of the analytes increases their hydrophilicity.

Although different HILIC LC–MS methods for quantitative analysis of a broad range of water soluble analytes in complex matrixes have been reported [41,42,43,44], we developed for the first time a ZIC-p-HILIC method using multivariate strategy to quantify the *dn*SP metabolites.

### 3.3. Linearity, Limits of Detection (LOD) and Limits of Quantification (LOQ)

Linearity, LOD and LOQ determinations were performed as already described in the experimental section. Appendix A compiles the results. The ZIC-p-HILIC-ESI-QTOF-MS method showed good linearity over the range of concentrations tested, with correlation coefficients (r^2^) that were greater than 0.990 for all of the analytes. The uracil base showed LOD values of 50 ng mL^−1^ and the amino acids between 800–820 ng mL^−1^; the LOQ values were of 100 ng mL^−1^ and 1300 ng mL^−1^, respectively. The LOD values of the nucleosides vary between 200–300 ng mL^−1^ and LOQ between 100–400 ng mL^−1^. The mono-, di-, and triphosphates nucleotides showed LOD values of 200 ng mL^−1^, 600 ng mL^−1^, and 2500 ng mL^−1^, respectively; and, LOQ of 600 ng mL^−1^, 200 ng mL^−1^, and 2000 ng mL^−1^, respectively. Finally, the LOD range were between 20–600 ng mL^−1^ and LOQ between 50–1600 ng mL^−1^ for the precursors and derivatives of the nucleosides.

### 3.4. Matrix Effect

The matrix effect was evaluated while using infected tomato leaf extract obtained as described in Section 2.4.3. Three different concentrations of each standard (low, medium, and high concentrations) were added to 50 mg L^−1^ of crude extract. Table 3 summarizes the results of the matrix effect that was evaluated at the mean concentration of the metabolites. We observed that all of the compounds except the nucleotide triphosphates had low matrix influence.

The *dn*SP metabolites concentrations were calculated from the slopes of the calibration curves. As seen in Table 3, the amino acids, nucleobase, nucleoside, and some derivatives were detected and quantified in the ZIC-p-HILIC-LC-MS analysis. Specifically, the mono-, di-, and triphosphate nucleotides were not determined (Appendix A).

## 4. Conclusions

It was possible to obtain a ZIC-p-HILIC method while using LC-ESI(−)−MS to evaluate the *dn*SP metabolites in tomato plants that were infected by *P. infestans* through the multivariate strategy of factorial design and response surface. The use of factorial planning allowed us to verify that the temperature, additive concentration and flow of the mobile phase were the parameters that mostly affect the separation and ionization of the metabolites. It was found that the elution gradient rate 2 min, pH 7.0, temperature 45 °C, mobile phase flow of 0.35 mL min^−1^, and additive (ammonium acetate) concentration of 6 mM are the optimal values for the chromatographic analysis, obtaining good results in the response variables: peak height (h), efficiency of the column (N), retention factor (*k*), and resolution (R). Additionally, the DoE allowed for the faster development of the method through the reduction of the total number of experiments. The method was developed for the analysis of *dn*SP metabolites in tomato plants, but it can be extrapolated to the analysis of these highly polar metabolites in a great diversity of complex matrices.

## Figures and Tables

**Figure 1 biomolecules-09-00328-f001:**
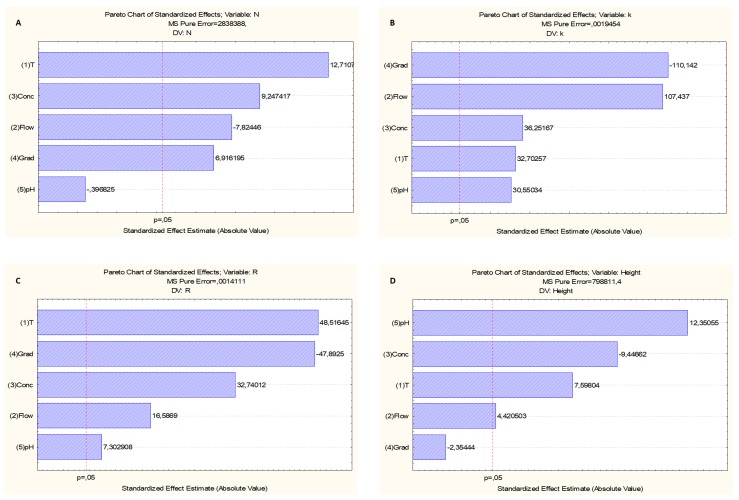
Pareto chart showing the standardized effect of independent variables and their interaction on the (**a**) Efficiency (N); (**b**) Retention factor (*k*); (**c**) Resolution (R); and, (**d**) Peak height (h). The length of each bar in the chart indicates the standardized effect of that factor on the response. The bar outside the reference line indicates that these terms contribute in the prediction of the dependent variables (N, *k*, R, and h). The negative coefficients for the model components indicate an unfavorable or antagonistic effect on the variables (N, *k*, R, and h), while the positive coefficients for the model components show a favorable or synergistic effect on the variables (N, *k*, R, and h).

**Figure 2 biomolecules-09-00328-f002:**
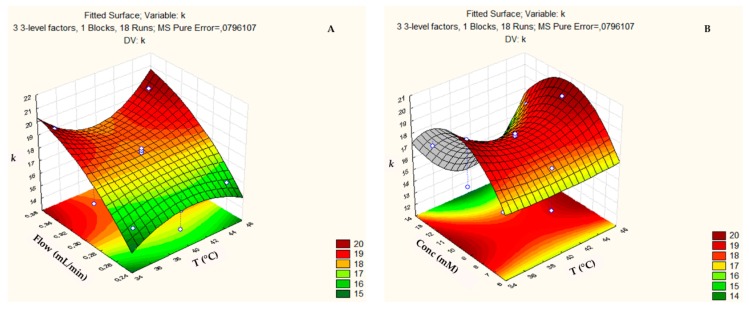
Three-dimensional response surface plot showing the effects of the mutual interactions between two independent variables (**a**) Flow and Temperature; (**b**) Concentration and Temperature and (**c**) Flow and Concentration on retention factor (*k*) of the chromatographic analysis on the ZIC-p-HILIC column.

**Figure 3 biomolecules-09-00328-f003:**
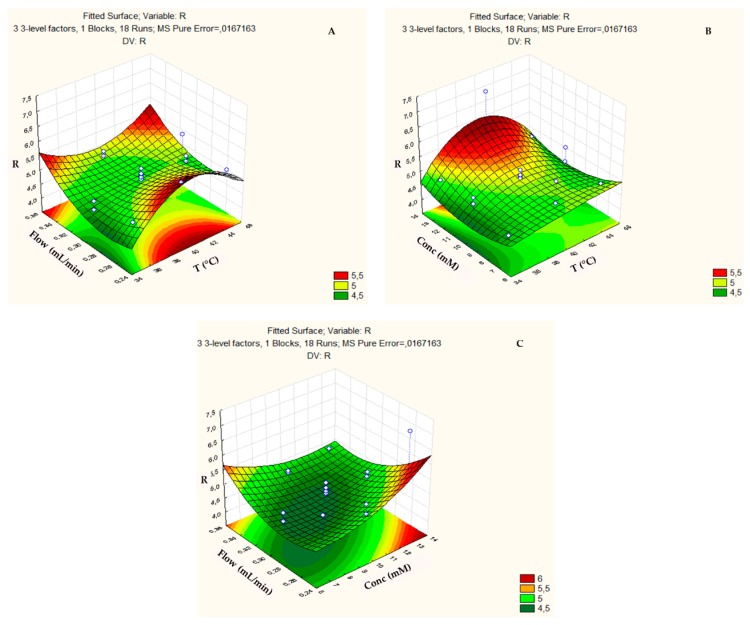
Three-dimensional response surface plot showing the effects of the mutual interactions between two independent variables (**a**) Flow and Temperature; (**b**) Concentration and Temperature; and, (**c**) Flow and Concentration on resolution (R) of the chromatographic analysis on the ZIC-p-HILIC column.

**Figure 4 biomolecules-09-00328-f004:**
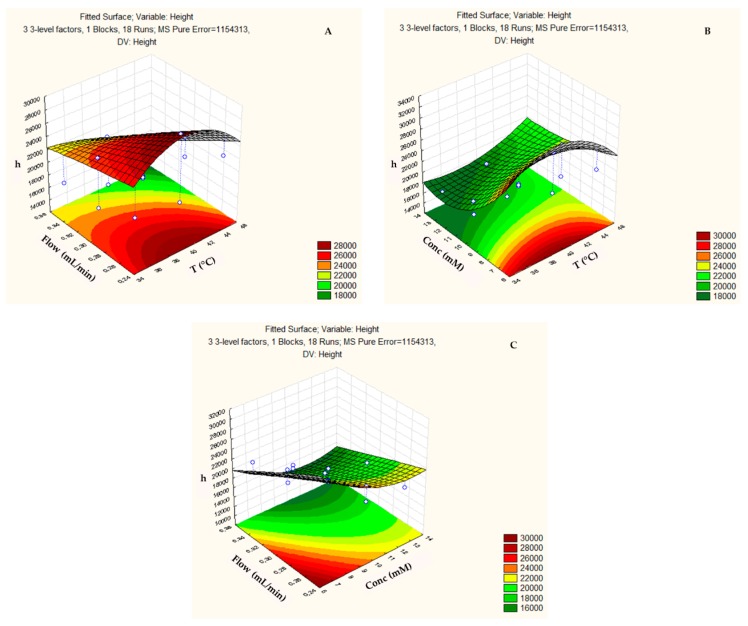
Three-dimensional response surface plot showing the effects of the mutual interactions between two independent variables (**a**) Flow and Temperature; (**b**) Concentration and Temperature; and, (**c**) Flow and Concentration on height (h) of the chromatographic analysis on the ZIC-p-HILIC column.

**Figure 5 biomolecules-09-00328-f005:**
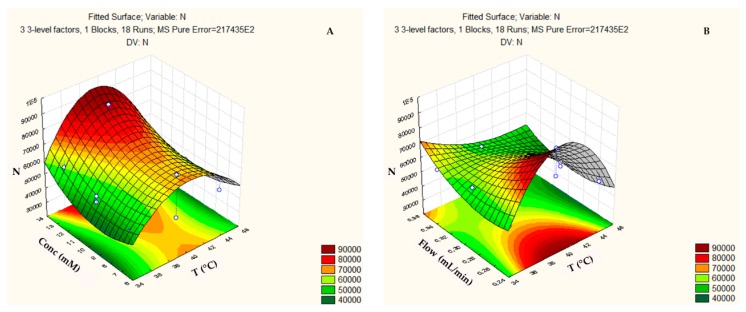
Three-dimensional response surface plot showing the effects of the mutual interactions between two independent variables (**a**) Flow and Temperature; (**b**) Concentration and Temperature; and, (**c**) Flow and Concentration on column efficiency (N) of the chromatographic analysis on the ZIC-p-HILIC column.

**Figure 6 biomolecules-09-00328-f006:**
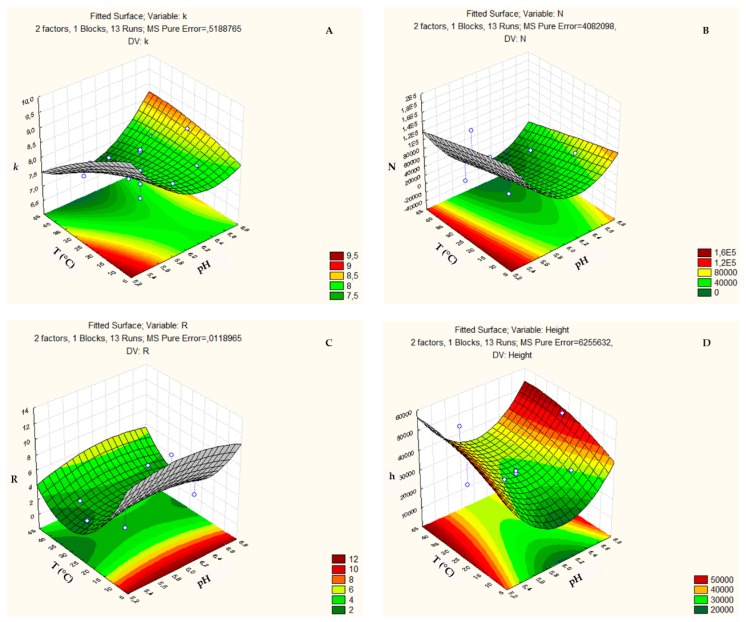
Three-dimensional response surface plot showing the effects of the mutual interactions between two independent variables pH and Temperature on (**a**) Retention factor (*k*); (**b**) Column efficiency (N); (**c**) Resolution (R); and, (**d**) Height (h) of the chromatographic analysis on the ZIC-HILIC column.

**Table 1 biomolecules-09-00328-t001:** Detailed description of the factorial designs for ZIC-p-Hydrophilic Interaction Liquid Chromatography (ZIC-p-HILIC) and ZIC-HILIC columns.

COLUMN (Design)	FACTOR (Abbreviation)	Minimum Level (−)	Medium Level (0)	Maximum Level (+)
ZIC-p-HILIC (Factorial #1)	Temperature in °C (T)	20	30	40
	Flow in mL min^−1^ (Flow)	0.15	0.2	0.25
	Additive concentration in mM (Conc)	5.0	7.5	10
	Gradient rate (Grad)	2	3	4
	pH	4	5.5	7.0
ZIC-HILIC (Factorial #2)	Temperature in °C (T)	25	40	55
	Flow in mL min^−1^ (Flow)	0.15	0.2	0.25
	Additive concentration in mM (Conc)	5	7.5	10
	Gradient rate (Grad)	2	3	4
	pH	3.0	4.5	6.0

**Table 2 biomolecules-09-00328-t002:** Detailed description of the Box-Behnken design and central composite design for ZIC-p-HILIC and ZIC-HILIC columns.

COLUMN (Design)	FACTOR (Abbreviation)	Minimum Level (−)	Medium Level (0)	Maximum Level (+)
ZIC-p-HILIC (Box-Behnken)	Temperature in °C (T)	35	40	45
	Flow in mL min^−1^ (Flow)	0.25	0.3	0.35
	Concentration in mM (Conc)	7.0	10	13
ZIC-HILIC (Central composite)	Temperature in °C (T)	15	20	35
	pH	5.5	6.0	6.6

**Table 3 biomolecules-09-00328-t003:** Matrix effect of the crude tomato infected leaves extract on the *dn*SP standards and metabolite concentration; (*n* = 3).

Analyte Concentration “Medium” (ng mL^−1^)	ng Analyte/mg of Tomato Leave Extract ^a^	ng Analyte/mg of Tomato Leave Extract ^b^
Aspartic acid (9000)	12,874.8 ± 219	11,265.5 ± 113
Glutamine (9000)	41,446.4 ± 630	40,882.4 ± 143
Uracil (2000)	11,598.6 ± 157	3202.3 ± 41
Uridine (2000)	24,078.3 ± 48	25,122.6 ± 201
Cytidine (3000)	8355.3 ± 74	4265.1 ± 189
UMP (4500)	2168.6 ± 53	ND
UDP-Glu (3000)	5720.7 ± 71	ND
UTP (16,000)	ND	ND
ATP (16,000)	ND	ND
CTP (16,000)	ND	ND
Carbamoyl aspartic acid (9000)	20,481.4 ± 190	ND
Orotic acid (2000)	387.1 ± 76	ND
DHO (2000)	4886.5 ± 84	1024.7 ± 18

^a^ Matrix effect using medium concentration of the standards; ^b^ Metabolite quantification without addition of standards; ND, Not determinate.

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
