# Peer review of "Hydrophilic Interaction Liquid Chromatography Coupled to Mass Spectrometry and Multivariate Analysis of the De Novo Pyrimidine Pathway Metabolites"

_biomolecules, 2019, doi:10.3390/biom9080328_

Reviewer 1 Report

It is a good paper and should be published. The authors present a new and useful methodology to optimize the ZIC-p-HILIC and LC-ESI (-)-MS detection and quantification of 13/14 metabolites of the important pyrimidine biosynthetic pathway (dnSP), using as their model tomato plants infected by P. infestans. Through a multivariate strategy of two-level factorial design and response surface analysis, the authors verified the most influential factors that affect the separation and ionization of the metabolites: temperature, additive concentration and mobile phase flow.  Optimized values were determined for the elution gradient rate, pH, temperature, mobile phase flow and additive (ammonium acetate) concentration for the chromatographic analysis, obtaining good results in the response variables: peak height, column efficiency, retention factor and resolution. Additionally, the design allowed for a faster development of the method by reducing the total number of experiments. This newly developed design will be of great benefit in the analysis of other similarly important metabolites in systems of high complexity.

Author Response

The reviewer has no comments

Reviewer 2 Report

My main concerns are novelty and appropriateness for Biomolecules. This is a highly methodological paper using  hydrophilic liquid chromatography coupled to mass spectrometry to separate and quantitate precursors or intermediates of the de novo pyrimidine biosynthesis pathway. The main innovation over the many existing methods is that two amino acids are quantitated (glutamine and aspartate) in addition to most intermediates of the route of UMP biosynthesis. By the way, one of the intermediates of this route, orotidine monophosphate, does not appear to have been monitored. Concerning appropriateness for Biomolecules, the text is highly involved, being understandable for people in the anlytical field, but much less accessible to a wider biomolecular community. Furthermore, it is of limited general interest, in particular the exploration of the variable to optimize the chromatogaphic procedure. I believe that it would be more appropriate for a chromatography-based journal such as one of the sections of the Journal of Chromatography. Cncerning the text itself, although well written, I miss a direct discussion comparing the present method with those reported previously for intermediates of the pyrimidine route.

Author Response

Reviewer 2

Comments and Suggestions for Authors

My main concerns are novelty and appropriateness for Biomolecules. This is a highly methodological paper using  hydrophilic liquid chromatography coupled to mass spectrometry to separate and quantitate precursors or intermediates of the de novo pyrimidine biosynthesis pathway. The main innovation over the many existing methods is that two amino acids are quantitated (glutamine and aspartate) in addition to most intermediates of the route of UMP biosynthesis. By the way, one of the intermediates of this route, orotidine monophosphate, does not appear to have been monitored. Concerning appropriateness for Biomolecules, the text is highly involved, being understandable for people in the anlytical field, but much less accessible to a wider biomolecular community. Furthermore, it is of limited general interest, in particular the exploration of the variable to optimize the chromatogaphic procedure. I believe that it would be more appropriate for a chromatography-based journal such as one of the sections of the Journal of Chromatography. Cncerning the text itself, although well written, I miss a direct discussion comparing the present method with those reported previously for intermediates of the pyrimidine route. 

Response to reviewer 2

The initial experimental design included the intermediate orotidine monophosphate. Due to the high cost of the standards, we prioritize dihydroorotate, orotate and UMP compounds which are highly significant of de novo pyrimidine pathway. As alternative, we thought about obtaining orotidine monophosphate through enzymatic synthesis using orotate, nonetheless, we did not get good results. Concerning to direct discussion comparing the present method with those reported previously for intermediates of the pyrimidine route, the LC-MS methods reported have focused on analyzing groups of metabolites related to the intermediaries of the route. That is, a pool of nucleotides, or nucleosides, or amino acids, or organic acids and so on. Nonetheless, a LC-MS method that involves the separation of de novo pyrimidine pathway metabolites as mentioned in the present research has not yet been reported. The following text will be included in the manuscript: Although different HILIC LC–MS methods for quantitative analysis of a broad range of water soluble analytes in complex matrixes have been reported [1–4], we developed for the first time and through multivariate strategy a ZIC-p-HILIC method to quantify the dnSP metabolites.

1.        Bajad, S.U.; Lu, W.; Kimball, E.H.; Yuan, J.; Peterson, C.; Rabinowitz, J.D. Separation and quantitation of water soluble cellular metabolites by hydrophilic interaction chromatography-tandem mass spectrometry. Journal of Chromatography A 2006, 1125, 76–88, doi:10.1016/j.chroma.2006.05.019.

2.        Teleki, A.; Sanchez-Kopper, A.; Takors, R. Alkaline conditions in hydrophilic interaction liquid chromatography for intracellular metabolite quantification using tandem mass spectrometry. Analytical Biochemistry 2015, 475, 4–13, doi:10.1016/j.ab.2015.01.002.

3.        Gika, H.G.; Theodoridis, G.A.; Vrhovsek, U.; Mattivi, F. Quantitative profiling of polar primary metabolites using hydrophilic interaction ultrahigh performance liquid chromatography–tandem mass spectrometry. Journal of Chromatography A 2012, 1259, 121–127, doi:10.1016/j.chroma.2012.02.010.

4.        Gallart-Ayala, H.; Konz, I.; Mehl, F.; Teav, T.; Oikonomidi, A.; Peyratout, G.; van der Velpen, V.; Popp, J.; Ivanisevic, J. A global HILIC-MS approach to measure polar human cerebrospinal fluid metabolome: Exploring gender-associated variation in a cohort of elderly cognitively healthy subjects. Analytica Chimica Acta 2018, 1037, 327–337, doi:https://doi.org/10.1016/j.aca.2018.04.002.

Reviewer 3 Report

The manuscript by Garcia et al describes the optimization of a LC-MS based method to analyze pyrimidine pathway related metabolites.

I am happy to conclude that the manuscript is very well written and very clear. It was my pleasure reading it. With that said I have hardly found anything that I think would significantly improve the quality of the manuscript. It is somehow lengthy, but in this case I find that good since it helps the reader. My concerns are mainly regarding figures:

Figure 1. Would it be possible to enlarge the text in the figure to make it more readable? 

Also, for all figures I would like a text (short summary) describing what the reader sees in the figure. This would help the readers.

Author Response

Reviewer 3

Comments and Suggestions for Authors

The manuscript by Garcia et al describes the optimization of a LC-MS based method to analyze pyrimidine pathway related metabolites.

I am happy to conclude that the manuscript is very well written and very clear. It was my pleasure reading it. With that said I have hardly found anything that I think would significantly improve the quality of the manuscript. It is somehow lengthy, but in this case I find that good since it helps the reader. My concerns are mainly regarding figures:

Figure 1. Would it be possible to enlarge the text in the figure to make it more readable? 

Also, for all figures I would like a text (short summary) describing what the reader sees in the figure. This would help the readers.

Response to reviewer 3

The legend of figures 1-6 was modified:

Figure 1. Pareto chart showing the standardized effect of independent variables and their interaction on the (a) Efficiency (N); (b) Retention factor (k); (c) Resolution (R) and (d) Peak height (h). The length of each bar in the chart indicates the standardized effect of that factor on the response. The bar outside the reference line indicates that these terms contribute in the prediction of the dependent variables (N, k, R and h). The negative coefficients for the model components indicate an unfavorable or antagonistic effect on the variables (N, k, R and h), while the positive coefficients for the model components show a favorable or synergistic effect on the variables (N, k, R and h).

Figure 2. Three-dimensional response surface plot showing the effects of the mutual interactions between two independent variables (a) Flow and Temperature; (b) Concentration and Temperature and (c) Flow and Concentration on retention factor (k) of the chromatographic analysis on ZIC-p-HILIC column.

Figure 3. Three-dimensional response surface plot showing the effects of the mutual interactions between two independent variables (a) Flow and Temperature; (b) Concentration and Temperature and (c) Flow and Concentration on resolution (R) of the chromatographic analysis on ZIC-p-HILIC column.

Figure 4. Three-dimensional response surface plot showing the effects of the mutual interactions between two independent variables (a) Flow and Temperature; (b) Concentration and Temperature and (c) Flow and Concentration on height (h) of the chromatographic analysis on ZIC-p-HILIC column.

Figure 5. Three-dimensional response surface plot showing the effects of the mutual interactions between two independent variables (a) Flow and Temperature; (b) Concentration and Temperature and (c) Flow and Concentration on column efficiency (N) of the chromatographic analysis on ZIC-p-HILIC column.

Figure 6. Three-dimensional response surface plot showing the effects of the mutual interactions between two independent variables pH and Temperature on (a) Retention factor (k); (b) Column efficiency (N); (c) Resolution (R) and (d) Height (h) of the chromatographic analysis on ZIC-HILIC column.

Round  2

Reviewer 2 Report

The authors have addressed the two specific queries that I had. Concerning appropriateness for the journal, if the journal editorial staff consider it appropriate I have nothing to say.